# Macrophage Migration Inhibitory Factor Suppresses Natural Killer Cell Response and Promotes Hypoimmunogenic Stem Cell Engraftment Following Spinal Cord Injury

**DOI:** 10.3390/biology14070791

**Published:** 2025-06-30

**Authors:** Shenglan Li, Yiyan Zheng, Haipeng Xue, Haiwei Zhang, Jiayun Wu, Xiaohui Chen, Miguel Perez Bouza, Samantha Yi, Hongxia Zhou, Xugang Xia, Xianmin Zeng, Qi Lin Cao, Ying Liu

**Affiliations:** 1The Vivian L. Smith Department of Neurosurgery, McGovern Medical School, The University of Texas Health Science Center at Houston, Houston, TX 77030, USA; shenglan.li@bcm.edu (S.L.); yiyzheng@fiu.edu (Y.Z.); haxue@fiu.edu (H.X.); nggaiwen@163.com (J.W.); 2Center for Stem Cell and Regenerative Medicine, The Brown Foundation Institute of Molecular Medicine, The University of Texas Health Science Center at Houston, Houston, TX 77030, USA; 3Center for Translational Science, Florida International University, 11350 SW Village Pkwy, Port St. Lucie, FL 34987, USA; haizhang@fiu.edu (H.Z.); xiachen@fiu.edu (X.C.); miguelpbouza@gmail.com (M.P.B.); yisam9829@gmail.com (S.Y.); hozhou@fiu.edu (H.Z.); xxia@fiu.edu (X.X.); 4Biomedical Science PhD Program, Herbert Wertheim College of Medicine, Florida International University, Miami, FL 33199, USA; 5Robert Stempel College of Public Health and Social Work, Florida International University, Miami, FL 33199, USA; 6RxCell Inc., Novato, CA 94945, USA; xzeng@xcellscience.com; 7Department of Physiology, Yong Loo Lin School of Medicine, National University of Singapore, Singapore 117593, Singapore

**Keywords:** universal stem cells, transplantation, immune rejection

## Abstract

Human induced pluripotent stem cells hold great promise for treating neurological diseases. One of the biggest challenges, however, is the immune system: if transplanted cells are not a perfect match, the body may reject them. To overcome this, we aimed to create “off-the-shelf”, universal cells that could be safely used in anyone, without needing a matched donor. Using CRISPR-mediated gene editing tool, we deleted two key genes, B2M and CIITA, that are responsible for making proteins recognized by the immune system. Additionally, we engineered the cells to produce MIF, which helps protect against natural killer cell attacks. Overall, our study shows that combining MIF overexpression with the removal of B2M and CIITA can produce universal cells that avoid rejection by the immune system. This approach could help make stem cell therapies more widely available and effective for spinal cord injuries and other diseases.

## 1. Introduction

Human induced pluripotent stem cells (hiPSCs) represent one of the most promising sources for cell-based therapies targeting a wide range of neurological disorders, including spinal cord injury (SCI) [1,2,3]. iPSCs offer the potential for personalized regenerative treatments by enabling the generation of patient-specific neural cells that are immunologically compatible. However, several critical challenges must be addressed before hiPSC-based therapies can be widely translated into clinical practice. These include the lengthy and labor-intensive process required to generate personalized hiPSC lines, the high cost associated with producing clinical grade hiPSCs for individual patients, and issues related to the scalability, standardization, and safety of the derived cell products. Overcoming these barriers will be essential to realize the full therapeutic potential of hiPSCs in treating SCI and other neurological conditions.

The development of universal iPSCs that possess hypoimmunogenic properties could help overcome many existing obstacles and significantly expedite the clinical translation of human iPSC-based therapies [4,5,6,7]. This is because hypoimmunogenic cells can be grafted allogeneically without eliciting immune rejection due to mismatched HLA types. Once generated, these cells can be differentiated into various functional cell types to meet the needs of different organ systems and disease conditions. After rigorous quality control, they can serve as universal, ready-to-use therapeutic products, much like an “off-the-shelf” drug, available for any patient in need of a transplant, without the need for personalized cell manufacturing each time. This feature is particularly advantageous for cell therapy in SCI, where early intervention during the acute phase is often more effective than treatment in the chronic stage. In such cases, there is no time to wait for the months- to year-long process required to generate, differentiate, and quality-control patient-specific iPSC-derived neural cells. Moreover, pursuing a personalized medicine approach would result in prohibitively high costs per patient, making a universal, hypoimmunogenic solution far more practical and scalable.

Strategies to manufacturing hypoimmunogenic iPSCs have been designed with knowledge learned from human placenta which is tolerated by the maternal immune system, and from numerous microorganisms that successfully evade human immune surveillance [4,8,9,10,11,12]. Current strategies to create hypoimmunogenic universal iPSCs for allogeneic therapy are focused on modifying MHC complexes, encoded by human leukocyte antigen or HLA genes [12,13,14,15,16,17,18,19,20], or immune cloaking by overexpressing multiple immunomodulatory proteins targeting various immune cells including T cells, NK cells, macrophages, monocytes, and dendritic cells [11,21].

In the context of the CNS, reports on the application of hypoimmunogenic cells remain limited. One study demonstrated that oligodendrocyte progenitor cells derived from a hypoimmunogenic hiPSC line, engineered via B2M and CIITA knockout alone, successfully restored myelination in a Canavan disease animal model, a condition marked by severe demyelination [20]. More recently, human embryonic stem cells engineered to overexpress eight immunomodulatory genes were differentiated into ventral midbrain dopaminergic neurons and transplanted into Parkinson’s disease animal models, showing promise for immune-compatible neural grafts [21].

Here, we manufactured a hypoimmunogenic line for universal use of allogeneic cell therapy by combining gene editing of HLA and overexpressing macrophage migration inhibitory factor (MIF), an immune modulatory gene. As proof of principle, we evaluated the in vivo survival and immune responses of our B2M/CIITA double KO and MIF-overexpressing hypoimmunogenic iPSC-derived NSCs in spinal cord injury animal models. This novel hybrid hypoimmunogenic cell strategy targets both the innate and adaptive immune systems through HLA gene modification and overexpression of immunomodulatory factors, offering the potential to enhance immune evasion. To our knowledge, this is the first study to apply hypoimmunogenic iPSC-derived neural lineage cells in SCI models, which may contribute to the future clinical translation of allogeneic iPSC-based therapies.

## 2. Materials and Methods

### 2.1. Cell Culture

All human iPSC lines were cultured in chemically defined mTeSR plus medium (Stem Cell Technologies, Vancouver, BC, Canada. Cat. No. 100-0276) in feeder-free conditions and passaged every 4~5 days with Accutase (Innovative Cell Technologies, San Diego, CA, USA. Cat. No. AT104) onto Matrigel (Corning, Corning, NY, USA. Cat. No. 08-774-552)-coated culture plates at a ratio of 1:4~1:8 with ROCK inhibitor Y-27632 (R&D Systems, Minneapolis, MN, USA. 10 µM) following manufacturer’s instructions, as described previously [22]. Human NK cell line NK-92MI (ATCC, Manassas, VA, USA. Cat. No. CRL2408) was cultured in high-glucose DMEM supplemented with 10% heat-inactivated fetal bovine serum, 1X Glutamax, and 1X non-essential amino acid (NEAA). All cell cultures were routinely tested for mycoplasma contamination on a monthly basis using a Mycoplasma PCR detection kit (ABM, Cat. No. G238).

### 2.2. Generation of HLA-KO and MIF-Overexpressing Hypoimmunogenic hiPSC Lines

The parental hiPSC line in this study is NCL2-GFP [23], a previously reported cGMP-grade hiPSC line that was further engineered to constitutively expresses GFP on chromosome 13 safe harbor locus [24,25]. This was also served as a control line in the current work. Next, both B2M and CIITA genes at the genomic level were knocked out by multiple single-guide RNAs (sgRNAs) targeted against B2M or CIITA via CRISPR/Cas9 gene editing. sgRNAs were designed, synthesized, and chemically modified by Synthego (Redwood City, CA, USA). sgRNA sequence and test primers are shown in Figure 1. Briefly, B2M-specific sgRNAs were delivered to the NCL2-GFP iPSCs via electroporation. Resulting B2M KO clones were verified by Sanger sequencing. Next, using B2M KO iPSC as the parental line, CIITA-specific sgRNAs were then electroporated into the B2M KO iPSCs to create a B2M and CIITA double KO iPSC clone. Null mutations for both genes were confirmed by Sanger sequencing (Figure 1) and the resulting clone was renamed as HLA-KO. To generate the MIF overexpressing HLA-KO iPSC line, the full-length MIF cDNA was obtained from Reprocell. The AAVS1 site-targeting vector and the corresponding transcription activator-like effector nucleases (TALENs) vector were a kind gift from Dr. Jizhong Zou at NIH [26]. Briefly, to make the AAVS1 MIF vector, MIF cDNA fragment connected to an EGFP cassette and a Blasticidin resistance cassette were digested and ligated into AAVS1-targeting vector backbone via Gibson assembly (New England Biolabs). Approximately 106 cells of HLA-KO hiPSCs were electroporated with 5 μg donor vector (CAG promoter-driven MIF expression cassette) and 2.5 μg of each TALEN, using Nucleofector (Lonza, Morristown, NJ, USA) Program A23. Cells were replated in mTeSR plus medium onto Matrigel. Cells were treated with 10 μM ROCK inhibitor for 24 h immediately after electroporation to enhance single-cell survival.

### 2.3. Differentiation of iPSCs into Neural Progenitor Cells (NPCs)

After being digested in 0.5 mM EDTA for 10 min in 37 °C, small clumps of human iPSCs were transferred to Petri dishes (Corning) and cultured in iPSC medium without basic fibroblast growth factor (bFGF) for 8 days to form embryoid bodies (EBs) as previously described [25,27]. EBs were then transferred to cell culture plates and continued to differentiate in neural induction medium (DMEM/F12 plus Glutamax, 1x non-essential amino acid (NEAA), 1x N2 supplement, and bFGF (20 ng/mL)). After differentiation for 2~3 days, neural rosettes were formed and manually isolated and dissociated into single cells by incubating in TrypLE for 5 min. The cells were expanded in Neurobasal medium supplemented with 1x NEAA, L-Glutamine (2 mM), 1x B27 supplement, and bFGF (20 ng/mL). All reagents were from ThermoFisher (Carlsbad, CA, USA) unless otherwise specified.

### 2.4. NK Cell LDH Cytotoxicity Assay

Neural progenitor cells (NPCs) derived from hiPSC lines NCL2-GFP (control), HLA-KO, and HLA-KO-MIF were used as target cells in the NK cell cytotoxicity assay. For LDH cytotoxicity NK cell killing assay, target cells were harvested and cell concentration was adjusted to 1 × 10^5^ cells/mL with culture medium. The concentration of effector cells (NK cells) was adjusted accordingly, and E/T ratios of 3:1, 1:1, 1:3 and 1:10 were tested. For LDH cytotoxicity assay, target cells and effector cells at different E/T ratios were co-incubated in 200 µL NK cell medium in a 96-well U bottom for 20 h. The supernatant was then collected and analyzed using the Pierce LDH cytotoxicity kit (Thermo Scientific, Waltham, MA, USA. cat. no. 88953). NK cell medium was set as the background control, NK cells (effector) or iPSC-derived NPCs (target cells) alone were used as controls for spontaneous release, and lysed target cells at the endpoint were taken as the maximum release. Absorbance at 490 nm and 680 nm was measured using a Bio-Rad plate reader (Hercules, CA, USA). To determine NK cell toxicity by measuring LDH release, the following formula was used: Cytotoxicity% = [(Experimental value-Effector cells spontaneous control−Target cells spontaneous control)/(Target cells maximal control−Target cells spontaneous control)] × 100.

### 2.5. NK Cell Degranulation Assay

To assess NK cell cytotoxic activity, a degranulation assay was performed by measuring NK cell surface expression of CD107a using flow cytometry. NK-92MI NK cell line was co-cultured with target cells (NPCs derived from hiPSCs lines NCL2-GFP control, HLA-KO, or HLA-KO-MIF) at E:T ratios of 3:1, 1:1, 1:3, or 1:10, in 96-well round-bottom plates. Target cells were then co-incubated with 3 × 10^5^ NK cells in NK cell medium supplemented with CD107a APC (Biolegend, San Diego, CA, USA. clone H4A3, cat. no. 328620) and protein transport inhibitor cocktail (Thermo Scientific, Waltham, MA, USA. cat. no. 00-4980-03) in the plate for 20 h. Cells were then harvested and stained with anti-human CD56 Brilliant Violet 421 conjugated antibody (Biolegend, San Diego, CA, USA. cat. no. 318327). Quantification of CD107a expression was counted using flow cytometry analysis on BD Aria II and analyzed using FlowJo software v11.

### 2.6. Transplantation of Hypoimmunogenic Cells to a Nude Rat Cervical SCI Model

All animal care and surgical interventions were undertaken in strict accordance with the approval of IACUC at the Florida International University. Surgical procedures were performed as described previously [3,28,29,30] with slight modifications. Briefly, anesthesia with ketamine-dexmedetomidine-butorphanol (K, 80 mg/kg; D, 0.25 mg/kg, B, 0.5 mg/kg) or isoflurane inhalation, adult NIH nude rats (3~6 months old, equal number of male and female animals, RNU rat, NIH-Foxn1rnu, Charles River) received a dorsal laminectomy at the 5th cervical vertebral level (C5) to expose the spinal cord and then a 150 kdyne moderate unilateral contusive injury was performed using an Infinite Horizons (IH) impactor (Infinite Horizons LLC, Lexington, KY, USA) with the spine stabilized using steel stabilizers inserted under the transverse processes one vertebra above and below the injury. After the contusion, the wound was sutured in layers and bacitracin ointment (Qualitest Pharmaceuticals, Huntsville, AL, USA) was applied to the wound area. Rats received 0.1 mL of gentamicin (2 mg/mL, s.c., ButlerSchein, Dublin, OH, USA) and then recovered on a water-circulating heating pad. The post-surgery care also included the treatment of analgesic agent, buprenorphine (0.05 mg/kg, s.c.; Reckitt Benckise, Hull, England), twice a day for three days with daily monitoring for 7 days. On Day 15 post-injury, rats with C5 unilateral contusion were randomly assigned to three groups, which received NCL2-GFP-NPCs, HLA-KO-NPCs, and HLA-KO-MIF-NPCs. There were 6 nude rats (3 male and 3 female) in each group. Two hours before transplantation, NPCs were detached from the dishes, collected by centrifugation at 1000× *g* for 4 min, and resuspended in 1 mL culture medium. Cell count and viability were assessed with trypan blue in a hemacytometer. The cell suspension was centrifuged a second time and resuspended in a smaller volume to give a density of 1 × 10^5^ viable cells/µL. Rats were re-anesthetized as above and the laminectomy site was re-exposed. Three injections were made at the injury center, 2 mm cranial and caudal to the injury center, respectively, at a depth of 1.1 mm and 0.6 mm laterally from the midline. At each site, 1 μL of cell suspension was injected through a glass micropipette with an outer diameter 50~70 μL and the tip sharp-beveled to 30–50° at rate of 0.2 μL/min as described previously [27,28,30]. Thus, a total of 300,000 cells were grafted into each injured spinal cord. The animals were allowed to survive 8 weeks after transplantation.

### 2.7. Transplantation of Hypoimmunogenic NPCs to a Humanized NSG Mouse SCI Model

Surgical procedures and cell grafting experiments were performed similarly to those described above in nude rats except for the following: adult (6–8 weeks old) humanized NSG mice Hu-NSG-MHC Class I/II KO-PBMC [31] (Jackson Laboratory, stock no. 742516, NOD.Cg-Prkdcscid H2-K1b-tm1Bpe H2-Ab1g7-em1Mvw H2-D1b-tm1Bpe Il2rgtm1Wjl/SzJ, homozygous for Prkdcscid, H2-K1tm1Bpe, H2-Ab1em1Mvw, H2-D1tm1Bpe, Il2rgtm1Wjl). They were anesthetized and received a dorsal laminectomy at the 10th thoracic vertebral level (T10) to expose the spinal cord and then a 60 kdyne moderate central contusive injury was performed with the spine stabilized using steel stabilizers inserted under the transverse processes one vertebra above and below the injury as described previously [27]. On Day 15 post-injury, mice with T10 unilateral contusion were randomly assigned to three groups, which received NCL2-GFP-NPCs, HLA-KO-NPCs, HLA-KO-MIF-NPCs. There were 6 mice in each group. Mice were re-anesthetized as above and a total of 300,000 cells were grafted into each injured spinal cord. The animals were allowed to survive 8 weeks after transplantation.

### 2.8. Quantitative RT-PCR

Total RNAs of iPSCs or iPSC-derived cells were extracted using Quick-RNA miniprep kits (Zymo Research, Irvine, CA, USA). One microgram of RNA was converted to cDNA using the iScript cDNA Synthesis Kit (Bio-Rad, Hercules, CA, USA). Quantitative RT-PCR (qPCR) was performed to quantify mRNA expression using the iQ SYBR Green Supermix kit (Bio-Rad). Expression of GAPDH was used as an internal control. The relative fold change in gene expression was evaluated using the comparative threshold cycle ΔΔCt method [32]. qPCR primers are listed in Appendix A.

### 2.9. Immunofluorescence and Immunohistochemistry

Immunofluorescence and immunohistochemistry were performed as previously described [3,27,33,34]. The undifferentiated iPSCs and differentiated neural lineage cells were characterized by immunocytochemistry. Briefly, cells grown on glass coverslips were fixed with 2% paraformaldehyde for 10 min at 4 °C and then washed 3 times in PBS. The fixed cells were incubated in blocking buffer (PBS containing 5% goat serum, 1% bovine serum albumin, and 0.1% Triton X-100) for 30 min at room temperature and then followed in blocking buffer containing primary antibodies at the indicated concentrations overnight at 4 °C. Appropriate secondary antibodies were used for single and double labeling. All secondary antibodies were tested for cross-reactivity and nonspecific immunoreactivity. For immunohistochemistry, animals were perfused transcardially with phosphate-buffered saline (PBS, pH 7.4), followed by 4% paraformaldehyde (PFA). The spinal cord segments were removed, post-fixed in 4% PFA for an additional 24 h, cryoprotected in 20% sucrose for 24 h, and subsequently in 30% sucrose for 24 h at 4 °C. Tissues were then embedded in O.C.T. compound (Fisher). Spinal cords were cryosectioned at 20 µm per slide either transversely or longitudinally. All slices were mounted serially on Superfrost Plus Gold Slides (Fisher). Slides were incubated in blocking buffer (10% donkey serum, 0.2% Triton X-100, in Tris-buffered saline) for 1 h at room temperature. The sections were then incubated in primary antibodies diluted in blocking buffer at 4 °C overnight. Secondary antibodies used are from Jackson Immuno Research: FITC-affiniPure Donkey anti-Rabbit, Cat. No. 711-095-152; Rhd-affiniPure Donkey anti-Rabbit, Cat. No. 711-025-152; Cy5-affiniPure Donkey anti-Rabbit, Cat. No. 711-175-152; Rhd-affiniPure Donkey anti-Mouse, Cat. No. 715-025-150; Cy5-affiniPure Donkey anti-Mouse, Cat. No. 715-175-150; AMCA-affiniPure Donkey anti-Rat, Cat. No. 712-155-153; Cy5-AffiniPure Donkey Anti-Rat IgG, Cat. No. 712-175-150. Appropriate secondary antibodies were used for single and double labeling. All secondary antibodies were tested for cross-reactivity and nonspecific immunoreactivity. The following primary antibodies were used: OCT 4 (1:500, Abcam, Waltham, MA, USA), SSEA4 (1:10, Development studies hybridoma Bank, DSHB, Iowa City, IA, USA), Nestin (1:200, Abcam), Sox2 (1:200, R&D Systems, Minneapolis, MN, USA), SOX1 (1:200, R&D Systems), β3 tubulin (1:4000, Sigma, Burlington, MA, USA), NeuN (1:1000, Sigma), MAP2 (1:500, Millipore, Burlington, MA, USA), NF160 (1:500, Sigma), neurofilament-light chain (NFL, 1:200, Sigma), SOX9 (1:200, Millipore), S100β (1:200, R&D Systems), GFAP (1:2000, Dako, Denmark), CD56 (1:200, Biolegend, San Diego, CA, USA), IBA1(1:200, Wako, Richmond, VA, USA). Ki-67 (Zymed, San Francisco, CA, USA. 1:200), and anti-human nuclei antigen (hN, 1:200, Millipore). Bis-benzamide (DAPI, 1:1000, Sigma) was used to visualize the nuclei. Images were captured using a Zeiss AxioVision microscope with z-stack split view function or Keyence BZ-X800 (Osaka, Japan).

### 2.10. Statistical Analysis

The difference in the number or percentage of positive cells of immunohistochemistry among the groups at specific survival times will be analyzed with one-way ANOVA followed by post hoc *t*-tests using a 95% confidence interval.

## 3. Results

### 3.1. Generation and Validation of HLA-KO, a Hypoimmunogenic Human iPSC Line

NCL2-GFP, a control iPSC line that constitutively expresses EGFP [23], was transfected with guide RNAs (gRNAs) designed to mutate B2M and CIITA genes by CRISPR-mediated non-homologous end joining (NHEJ). According to the genomic sequence of B2M (transcript ID ENST00000648006.3) and CIITA (transcript ID ENST00000324288.12), three gRNAs for each gene were designed (Appendix A). gRNAs for B2M were delivered first to create four B2M KO clones via CRISPR-mediated NHEJ, and then gRNAs for knocking out CIITA were electroporated to the B2M single KO clone to generate a double KO line (renamed as HLA-KO), which was verified by PCR and Sanger sequencing. As shown in Figure 1A, multiple insertions, deletions, or mutations in B2M and CIITA genes can be found in a representative clone of the HLA-KO iPSC line compared to the parental iPSC control (NCL2-GFP), indicating successful biallelic KO of both genes. To further confirm B2M/CIITA double KO at the protein level and to examine the effects of B2M/CIITA double KO in the expression of HLA class I and class II complexes, we performed immunocytochemistry staining for B2M, CIITA, HLA-I, and HLA-II. As shown in Figure 1B–E, the control NCL2-GFP line exhibited robust expression of B2M and HLA-I (Figure 1B,C) alongside with GFP, a built-in constitutive fluorescent marker. In contrast, the HLA-KO line showed a complete lack of expression of B2M and HLA-I (Figure 1B,C). Upon 100 ng/mL IFNγ treatment for 48 h, expression of CIITA and HLA-II could be detected in NCL2-GFP (Figure 1D,E). However, the HLA-KO cells did not express CIITA and HLA-II (Figure 1D,E). Thus, our results show that a hypoimmunogenic HLA-KO hiPSC line was successfully generated by CRISPR-mediated deletion of both B2M and CIITA.

### 3.2. The Properties and Neural Differentiation of HLA-KO iPSCs In Vitro

To examine the potential effects of B2M/CIITA double KO on the important properties of hiPSCs, we examined the karyotype, short tandem repeat (STR) profile, and pluripotency markers in HLA-KO. As shown in Figure 2A, the HLA-KO clone maintained a normal karyotype, the same as the parental NCL2-GFP (Appendix A). Similarly, STR analysis also matched the parental control line (Figure 2B). Furthermore, the HLA-KO iPSCs showed typical iPSC morphology by forming compact colonies with well-defined, smooth edges, which uniformly expressed pluripotency markers SSEA4, NANOG, and OCT4 (Figure 2C–F). Our results show that B2M/CIITA double KO does not change the properties of hiPSCs, such as karyotype, STR, or pluripotency.

We next examined whether B2M/CIITA double KO affected the neural differentiation of hiPSC. Compared to the parental NCL2-GFP line, HLA-KO iPSCs showed similar neural differentiation capability and pattern in vitro, in that they efficiently generated neural rosettes and gave rise to neural progenitor cells (NPCs) expressing typical markers such as SOX1, SOX2, and NESTIN (Figure 2G–I). In neuronal differentiation conditions, NPCs derived from HLA-KO hiPSCs differentiated into mature neurons expressing βIII tubulin (TuJ1), MAP2, NeuN, Synaptophysin, and NF160 (Figure 2J–N). Furthermore, the progenitor cells can differentiate into astrocytes expressing GFAP in the astrocyte differentiation condition (Figure 2O). These results indicate that B2M/CIITA double KO does not affect neural differentiation in vitro in the period tested in this study.

### 3.3. Generation of MIF-Overexpressing HLA-KO iPSC to Optimize Hypoimmunogenic Properties

Previous studies showed that a complete B2M KO, which would eliminate all HLA class I molecules, including HLA-E and HLA-G, was associated with the increasing risk of NK cell-mediated graft rejection due to the absence of these inhibitory signals [13,35,36,37,38]. Macrophage migration inhibitory factor (MIF) is a cytokine that can downregulate NKG2D expression on NK cells and has been suggested to suppress NK responses upon grafting [39,40]. To mitigate the potential consequences of HLA-KO in NK cell-mediated graft rejection, we implemented a strategy to reduce host NK cell activation by overexpressing MIF. We therefore designed a vector to constitutively express MIF, driven by a CAG promoter, at AAVS1, a safe-harbor locus located at chromosome 19 (Figure 3A,B). The resultant clone was renamed HLA-KO-MIF, and its overexpression of MIF was verified by qPCR (Figure 3C).

### 3.4. NK Cell Cytotoxicity and Degranulation Are Significantly Reduced When Co-Incubated with MIF-Overexpressing iPSC-Derived NPCs

To assess the NK cell response to the hypoimmunogenic cells generated in this study, we performed NK cell cytotoxicity and degranulation assays in vitro. Compared to the NCL2-GFP control, the HLA-KO cells exhibited a similar or slightly increased capacity to trigger NK cell response in cytotoxicity assays, depending on effector-to-target ratios (E/T, where effectors are NK cells, and targets are iPSC-derived NPCs, Figure 4). Interestingly, NK cell responses in co-incubation with HLA-KO-MIF NPCs were significantly diminished in conditions with all three E/T ratios tested in the LDH cytotoxicity assay. Consistently, the degranulation assay showed that co-incubation of HLA-KO cells with NK cells resulted in significantly higher CD56+/CD107a+ expression (mean value: 22.16) compared to NCL2-GFP control target cells (mean value: 19.64, *p* = 0.0318), indicating that the double KO of B2M and CIITA significantly triggered CD107a surface expression in CD56+ NK cells. Notably, NK cell responses were significantly reduced when HLA-KO-MIF cells were used as target cells, with mean CD56+/CD107a+ expression decreasing from 22.16 in HLA-KO to 17.48 in HLA-KO-MIF (*p* = 0.0019). Collectively, these data demonstrate that MIF overexpression attenuates NK cell responses in both LDH cytotoxicity and degranulation assays, suggesting a potential role for MIF in suppressing NK cell activity.

### 3.5. Overexpression of MIF Reduces Host NK Cell-Mediated Immune Responses

To directly test the effects of MIF on NK cell-mediated rejection in vivo, we examined the activation of NK cells upon grafting to athymic T-cell-deficient nude rats. Since immune rejection of grafts is primarily mediated by T cells and NK cells, host NK cells become the predominant drivers of rejection in T-cell-deficient recipients. At 8 weeks post grafting, inflammatory cells (i.e., IBA1+ activated macrophages or CD56+ NK cells) clustered at the injury epicenter, the areas immediately adjacent to the spared neural tissue, where the grafted human cells were localized (Figure 5). These cells showed the typical amoeboid morphology. In the NCL2-GFP control group (Figure 5A–E), cell surface contacts between the activated inflammatory cells and the grafted GFP+ human cells were only occasionally observed. However, in HLA-KO iPSC-derived NPC-grafted animals (Figure 5F–J), the inflammatory cells often made close contact with the grafted cells and were seen engulfing grafted human cells, leaving cell debris nearby. These data suggest that HLA KO increases NK cell-mediated rejection of grafted cells. In the HLA-KO-MIF group (Figure 5K–O), the number of NK cells among the grafts decreased and NK cell engulfment appeared compromised. The survival of grafted NPCs is increased. These data indicate that MIF overexpression in HLA-KO-MIF iPSC-derived NPC grafts could evade host NK cell-mediated attack.

### 3.6. Expression of MIF Promotes the Survival of Grafted Neural Progenitor Cells in T-Cell-Deficient Nude Rats

We first quantified the survival of grafted NPCs derived from control, HLA-KO, and HLA-KO-MIF hiPSCs upon grafting to nude rats. At 2 weeks post grafting (Figure 6), the number of surviving grafted NPCs derived from HLA-KO hiPSCs was significantly lower than that of NPCs derived from control hiPSCs while the number of surviving grafted NPCs in the HLA-KO-MIF group was significantly increased compared to those derived from HLA-KO hiPSCs, confirming that MIF overexpression can at least partially rescue the poor survival of HLA-KO NPCs upon grafting. We further examined whether overexpression of MIF will affect the differentiation of grafted NPCs in vivo. As shown in Figure 7, compared to NCL2-GFP control and HLA-KO iPSC-derived NPCs, HLA-KO-MIF NPCs showed similar proliferation capacity as indicated by Ki-67 staining, and differentiation potential into neuronal (Figure 7R) and astrocytic (Figure 7S) lineages at eight weeks post grafting.

### 3.7. Combination of HLA-KO and MIF Expression Increased Survival and Migration of Grafted hiPSC-Derived NPCs in Humanized NSG Mice

To further evaluate the immune response of our hypoimmunogenic strategy in the injured environment in vivo with a human immune system, we grafted iPSC-derived NPCs to the injured spinal cords of humanized NSG mice, whose mouse immune system was wiped out and replaced with human immune cells (Figure 8 and Figure 9). Some NPCs derived from NCL2-GFP control hiPSCs survived in the injured spinal cord at 3 days post grafting, but their numbers significantly declined by 14 days. Similarly, grafted NPCs derived from HLA-KO hiPSCs were present at 3 days post grafting but showed poor survival by day 14. The number of surviving HLA-KO NPCs was significantly lower than that of control NPCs, suggesting that HLA knockout alone is insufficient to prevent human immune cell-mediated rejection in humanized mice. In contrast, the HLA-KO-MIF group exhibited significantly improved NPC survival and migration at both 3 and 14 days post grafting compared to NPCs derived from either control or HLA-KO hiPSCs (Figure 8). In addition, overexpression of MIF does not affect neuronal differentiation (Figure 9), although only a few GFP+ human cells started to express NeuN, an early neuron marker, at 14 d post graft. Similarly, overexpression of MIF does not seem to alter glial differentiation, as SOX9+ astrocyte progenitors were observed to have derived from both control and HLA-KO-MIF iPSC-derived NPCs (Figure 9).

## 4. Discussion

The goal of this study was to generate a hypoimmunogenic hiPSC line and examine its immunogenicity, survival, differentiation, and integration in SCI animal models. We first generated HLA-KO, by knocking out both B2M and CIITA genes in NCL2-GFP, a current Good Manufacturing Practices (cGMP)-grade parental hiPSC line, using CRISPR-mediated gene editing. We found that compared to NCL2-GFP, HLA-KO iPSC-derived NPCs had increased NK cell-mediated cytotoxicity in vitro and showed poor survival in vivo when grafted into the athymic T-cell-deficient nude rats, which was likely caused by the loss of functional HLA-E and HLA-G, both of which are critical for inducing NK cell tolerance. To overcome this issue, we overexpressed MIF, by inserting a constitutively active MIF expression cassette into the AAVS1 safe harbor locus on chromosome 19 in HLA-KO iPSCs. We characterized HLA-KO-MIF hiPSCs with in vitro neural differentiation and NK cell cytotoxicity assays. Our data showed that MIF overexpression significantly reduced NK cell-mediated cytotoxicity. Importantly, when grafted into the injured spinal cords of T-cell-deficient nude rats, HLA-KO-MIF iPSC-derived NPCs showed improved survival without impairing their differentiation or migration potential. Furthermore, in a humanized mouse SCI model, HLA-KO-MIF NPC grafts exhibited enhanced survival and protection from NK cell-mediated rejection. Thus, HLA-KO-MIF represents a hypoimmunogenic, universal hiPSC line suitable for SCI cell therapy.

Given previous reports that B2M and CIITA double KO alone can confer human pluripotent stem cells (hPSCs, a collective term referring to both hESCs and hiPSCs) hypoimmunogenic in specific contexts, such as retinal pigment epithelium (RPE) cells [16] and oligodendrocyte progenitor cells (OPCs) [20], we initially tested HLA-KO-only hiPSCs in our SCI study. However, the success of the double KO strategy in RPE and OPC applications may be attributed to the immune-privileged nature of the target sites or to the high endogenous expression of immunomodulatory molecules. For instance, OPCs derived from B2M/CIITA KO iPSCs were shown to express high levels of CD47, a “do-not-eat-me” signal that inhibits phagocytosis by monocytes and macrophages, potentially facilitating their integration into the demyelinating brain of the Canavan disease [20]. In contrast, the HLA-KO NPCs used in our SCI model did not show elevated CD47 expression. Moreover, the immune environment of the spinal cord injury may involve stronger NK cell responses compared to macrophage- or monocyte-mediated effects in the Canavan disease model, possibly explaining the insufficient immune evasion observed with the double KO line in our setting. Therefore, additional modifications were necessary to improve the hypoimmunogenicity of our HLA-KO iPSC line.

To restore NK cell tolerance in HLA-KO cells, at least two strategies are possible: knock-in of HLA-E and/or HLA-G, or overexpression of immunomodulatory factors that suppress NK cell-mediated responses. We chose the latter approach to avoid additional rounds of CRISPR gene editing and minimize the risk of off-target effects. Additionally, if successful, this strategy could be more clinically translatable, as cytokine overexpression can potentially be replaced by local administration of recombinant proteins or small molecules of similar functions, bypassing the need for extensive genetic modification.

Among several candidates of immunomodulatory factors, we selected MIF for the following reasons: (1) MIF has been shown to transcriptionally downregulate NKG2D, an activating receptor on NK cells that promotes cytotoxicity [40], thereby acting as a negative NK cell checkpoint ligand for NK cell activation. (2) As a secreted cytokine involved in both innate and adaptive immune responses, MIF also skews macrophages toward an anti-inflammatory phenotype during phagocytosis [41,42,43], which may further help reduce graft rejection. (3) MIF is endogenously expressed in multiple regions of the developing brain and spinal cord, where it plays a neuroprotective role by promoting neuronal differentiation and neurite outgrowth [41,42,44,45,46,47,48] by enhancing neuronal differentiation and increasing neurite outgrowth [49,50,51,52]. Loss-of-function studies have shown that MIF deficiency leads to reduced expression of neuronal markers such as PSA-NCAM and doublecortin (DCX), and pharmacologic inhibition of MIF impairs neuronal proliferation and neurite extension in the hippocampus [53,54]. Indeed, our results support MIF as a promising candidate cytokine to mitigate NK cell-mediated rejection without impairing graft survival or neural differentiation in the SCI animal hosts.

To comprehensively evaluate our hypoimmunogenic line, we tested HLA-KO-MIF iPSC-derived NPCs by grafting them into two SCI models: T-cell-deficient nude rats and humanized NSG mice. The nude rat model has been established to evaluate human NPC grafts [1,2,3]. Due to its lack of T cells while retaining functional NK cells, the nude rat model is specifically used to examine whether the complete absence of the HLA class I complex (including NK tolerance molecules HLA-E and HLA-G) in HLA-KO iPSC-derived NPCs elicits an increased NK cell-mediated attack on grafts, and if so, to test potential strategies to mitigate this response. Our data support that the HLA-KO grafts trigger heightened NK cell attack from the host, which can be mitigated by overexpression of MIF, a cytokine that suppresses NK cell activity, possibly through inhibition of the NKG2D receptor.

The second SCI model used in this study is the spinal cord contusion model in humanized NSG mice. These mice are engineered to possess a functional human immune system, making them an ideal platform to assess whether our hypoimmunogenic iPSC-derived NPCs truly exhibit reduced immunogenicity. Our data support the hypothesis that compared to NCL2-GFP control and HLA-KO grafts, HLA-KO-MIF iPSC-derived NPC grafts could survive better, and migrate and differentiate along neural lineages, in injured humanized NSG mouse spinal cord. However, this injury model presents an inherent limitation. Although various humanized NSG mouse strains are commercially available, their typical lifespan of 1–3 months is not optimal for use as hosts in SCI grafting studies, for several reasons: (1) Contusion injury is generally induced in mice aged 6–8 weeks, with cell grafting performed 1–2 weeks post-injury. Proper evaluation of neural differentiation and behavioral outcomes requires an additional 2–3 months, meaning a minimum lifespan of 5–6 months is necessary. (2) For long-term studies assessing graft maturation and terminal differentiation, potential graft overgrowth and long-term safety, and sustained behavioral outcomes and functional recovery, animals must be maintained for at least another 8–12 months. Therefore, the development of a more robust and long-lived humanized mouse model is crucial for the future evaluation of hypoimmunogenic cells in SCI research.

## 5. Conclusions

Clinical trials using neural stem cells (NSCs) derived from fetal brains for treating SCI have been suspended due to lack of efficacy and the side effects related to immune suppression. Universal, hypoimmunogenic iPSC-derived NPCs could be a better cell source for SCI therapy in that they could be readily produced homogenously in bulk and omit the need for an immune-suppressive regimen. Our work established HLA-KO iPSCs derived from a cGMP-grade human iPSC line. Using HLA-KO as the parental line, we further engineered and validated an HLA-KO-MIF hypoimmunogenic iPSC line in SCI animal models. Our data provide feasibility of hypoimmunogenic iPSCs in SCI treatment. We aim to accelerate the translation of iPSC allogeneic cell therapy to the clinics, which could lead to critical information in developing safe and effective iPSC-based restorative therapies for CNS injury and other neurological conditions in the future.

## Figures and Tables

**Figure 1 biology-14-00791-f001:**
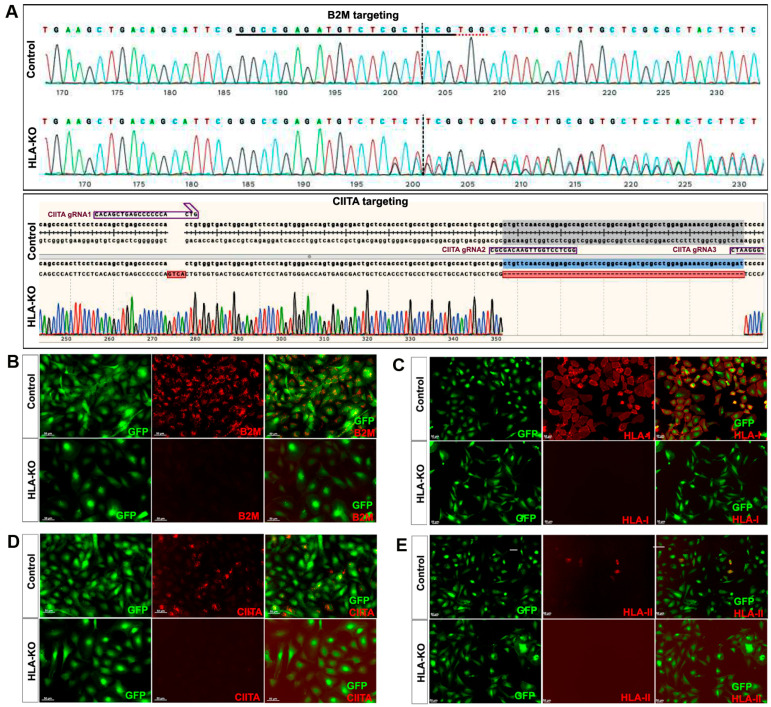
Creation and validation of HLA-KO iPSCs. (**A**) NCL2-GFP, a control iPSC line that constitutively expresses GFP, was transfected with guide RNAs designed to mutate B2M and CIITA genes by CRISPR-mediated non-homologous end joining. Sanger sequencing of a representative clone shows multiple insertions, deletions, or mutations in B2M and CIITA genes. The double KO clone is thus renamed HLA-KO. (**B**–**E**) Cells were treated with IFNγ (100 ng/mL) for 48 h. Expression of B2M, HLA class I, CIITA, and HLA class II molecules was assessed by immunocytochemistry. While expression of these markers was detected in NCL2-GFP control cells, HLA-KO iPSCs showed a complete absence of expression. All cultured cells are delineated by the built-in constitutive GFP expression. Bar, 100 µm. For complete information on guide RNA sequences (B2M gRNA1, 2, 3 and CIITA gRNA1, 2, 3) and primer sequences, please refer to Appendix A.

**Figure 2 biology-14-00791-f002:**
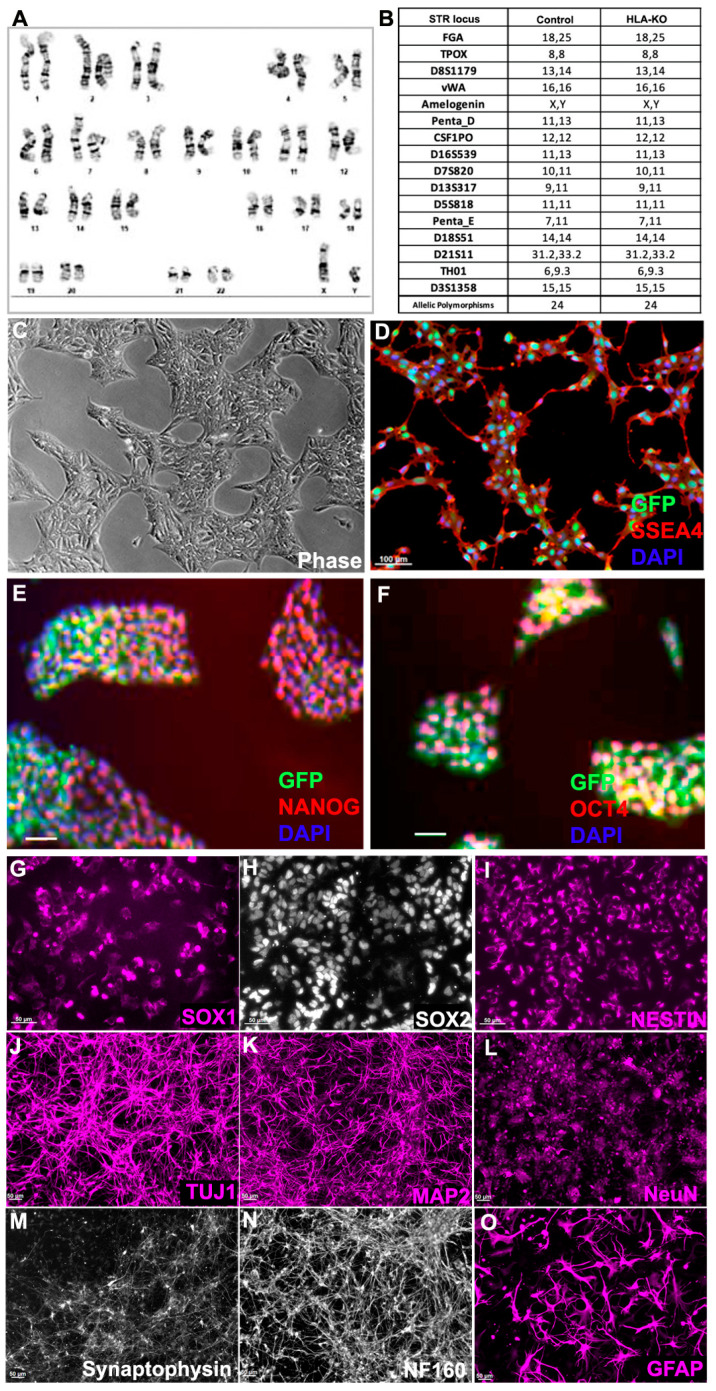
HLA-KO iPSCs can be differentiated in vitro into cells of the neural lineage. (**A**) HLA-KO maintained a normal karyotype. (**B**) Short tandem repeat analysis showed identical profile of HLA-KO and its parental control. (**C**–**F**) HLA-KO expressed pluripotency markers SSEA4, NANOG, and OCT4. (**G**–**O**) HLA-KO iPSCs were directly induced toward the neural lineage and gave rise to SOX1+, SOX2+, and NESTIN+ NPCs (**G**–**I**), neurons, labeled by TuJ1, MAP2, NeuN, Synaptophysin, and NF160 (**J**–**N**), as well as GFAP+ astrocytes (**O**). Bar, 100 µm (**C**–**F**), 50 µm (**G**–**O**).

**Figure 3 biology-14-00791-f003:**
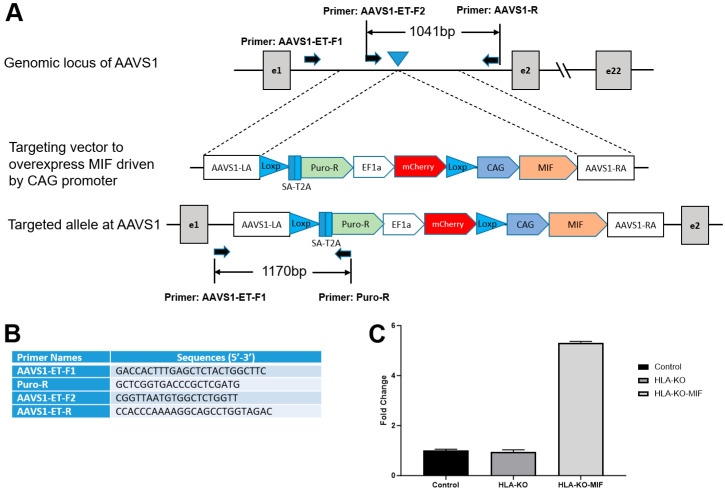
Vector design to overexpress MIF at AAVS1 site in HLA-KO to generate HLA-KO-MIF. (**A**) Vector design to target MIF expression cassette to the safe harbor locus AAVS1 on chr. 19. (**B**) Primers to test and verify correctly targeted human iPSC clones. (**C**) Quantitative PCR (qPCR) confirmed overexpression of MIF in HLA-KO-MIF iPSCs. Data are mean ± SEM. *n* = 3. Student’s *t*-test.

**Figure 4 biology-14-00791-f004:**
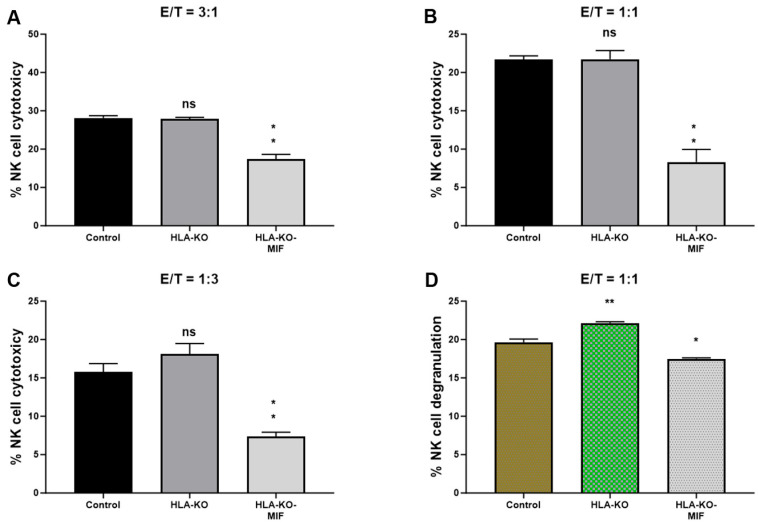
NK cell responses are diminished in MIF-overexpressing NPCs. (**A**–**C**) NK cell cytotoxicity assays were performed with bars representing the percentage of cytotoxicity against NPCs derived from (1) NCL2-GFP control, (2) HLA-KO, and (3) HLA-KO-MIF iPSCs. Similar trends were observed across different effector/target (E/T) ratios. For E/T ratio of 3:1, the mean % of NK cell cytotoxicity was HLA-KO (27.88) vs. HLA-KO-MIF (17.38), *p* = 0.0013; for E/T ratio of 1:1, the mean % of cytotoxicity was HLA-KO (21.74) vs. HLA-KO-MIF (8.308), *p* = 0.0026; and for E/T ratio of 1:3, the mean % of cytotoxicity was HLA-KO (18.08) vs. HLA-KO-MIF (7.385), *p* = 0.0021. (**D**) NK cell degranulation was assessed by measuring the percentage of CD107a+ cells among total CD56+ NK cells, indicating the level of NK cell activation induced by NPCs from the three groups. Data are mean ± SEM. *n* = 3. Student’s *t*-test. * *p* < 0.05; ** *p* < 0.01. ns, not significant.

**Figure 5 biology-14-00791-f005:**
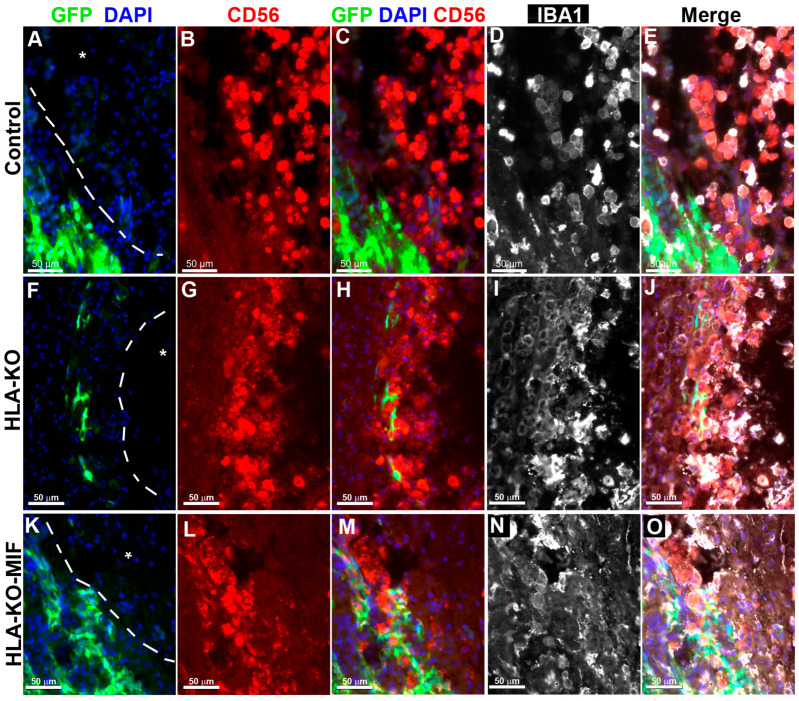
Inflammatory cells cluster at the injury site and interact with grafted cells**.** Eight weeks post grafting to the injured spinal cords of athymic homozygous T-cell-deficient Foxn1^RNU^-nude rats, clusters of CD56+ NK cells and/or IBA1+ activated macrophages remained adjacent to the injury sites (delineated by dotted lines and an asterisk). In the NCL2-GFP control group (**A**–**E**), interactions between inflammatory and grafted cells were only occasionally observed. In contrast, in HLA-KO iPSC-derived NPCs grafted animals (**F**–**J**), the inflammatory cells (labeled by CD56+ and/or IBA1+) could be seen engulfing grafted cells with visible cell debris. In the HLA-KO-MIF group (**K**–**O**), the engulfment was decreased. Bar, 50 µm.

**Figure 6 biology-14-00791-f006:**
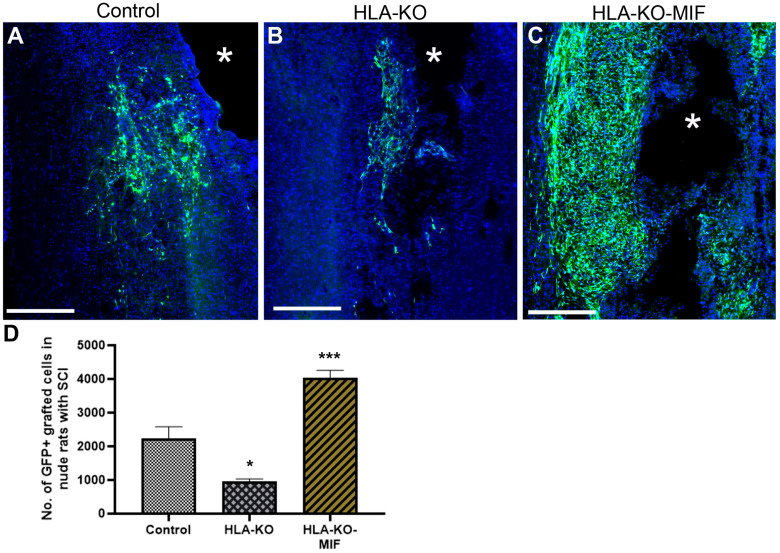
Enhanced survival of NPCs overexpressing MIF upon grafting in T-cell-deficient rats. Compared to the NCL2-GFP control (**A**) and HLA-KO (**B**), NPCs derived from HLA-KO-MIF (**C**) iPSCs exhibited significantly improved survival in nude rat spinal cords 14 days after transplantation. The number of GFP+ cells per spinal cord were quantified from a total of 4 mm long fragment of spinal cord tissue (2 mm above and 2 mm below the injury site) at 14-days post grafting (**D**). The asterisk (*) indicates injury site. Data are mean ± SEM. *n* = 3. Student’s *t*-test. * *p* < 0.05; *** *p* < 0.001. Bar, 400 µm.

**Figure 7 biology-14-00791-f007:**
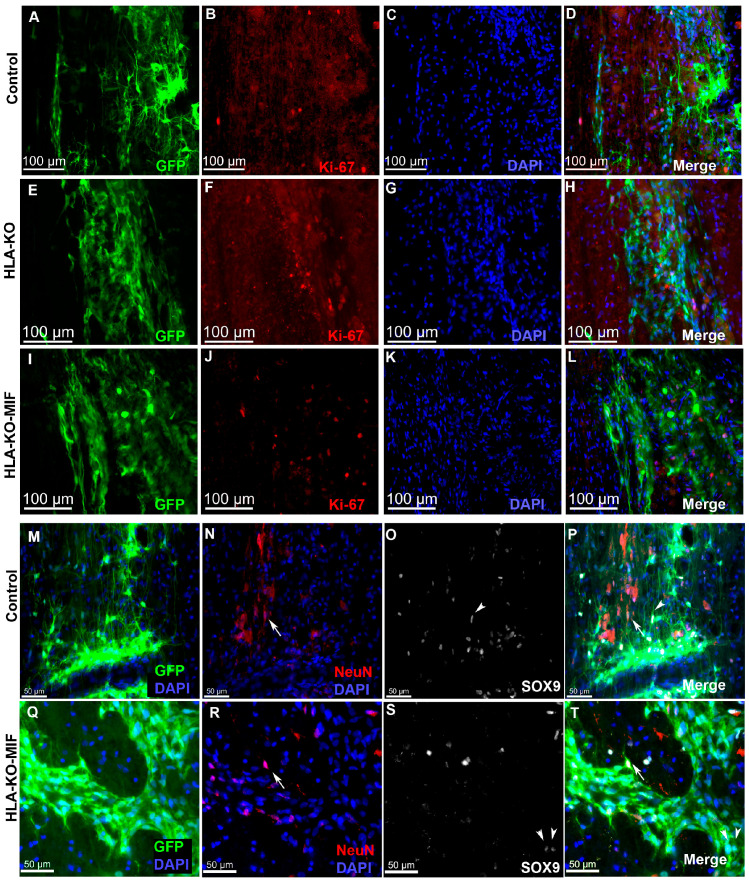
HLA-KO-MIF NPCs and control NPCs show similar proliferation and neural differentiation properties upon grafting to nude rat spinal cords. An equal number of NPCs derived from NCL2-GFP, HLA-KO, and HLA-KO-MIF iPSCs were grafted to the injured spinal cord (at C5) of T-cell-deficient nude rats. Eight weeks post grafting, proliferation of the grafted cells was evaluated with Ki-67 staining (**A**–**L**) and neural differentiation was evaluated with NeuN and SOX9 staining (**M**–**T**) at the grafting location. The number of NeuN+ neurons ((**N**,**P**,**R**,**T**), arrows), and the number of SOX9+ astrocyte progenitors ((**O**,**P**,**S**,**T**), arrowheads) were also similar in the control and HLA-KO-MIF groups. Bar, 100 µm in (**A**–**L**); 50 µm in (**M**–**T**).

**Figure 8 biology-14-00791-f008:**
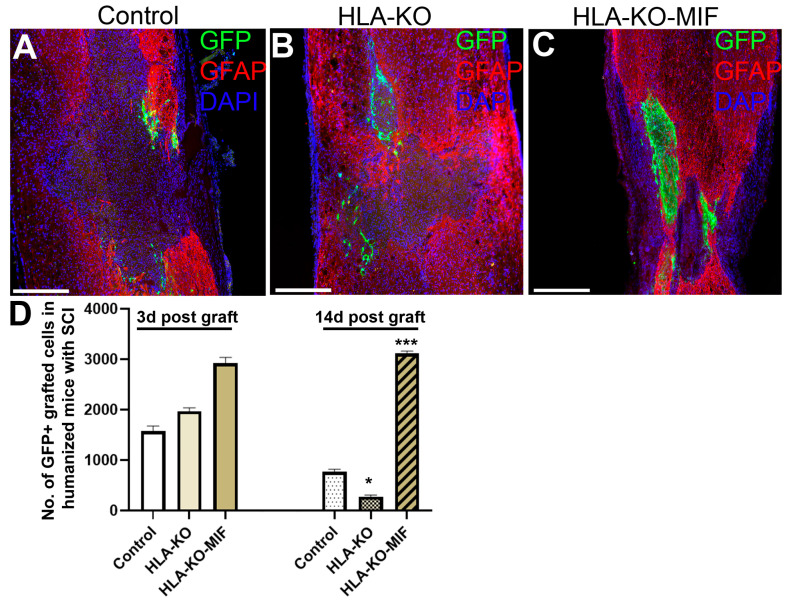
Overexpression of MIF enhances survival of grafted cells upon grafting to humanized NSG mice. Compared to the NCL2-GFP control (**A**) and HLA-KO (**B**), NPCs derived from HLA-KO-MIF (**C**), iPSCs exhibited significantly improved survival in the injured spinal cord of humanized NSG mice 14 days after transplantation (**A**–**C**). The number of GFP+ cells per spinal cord were quantified from a total of 4 mm long fragment of spinal cord tissue (2 mm above and 2 mm below the injury site) at 3 and 14 days post grafting to injured spinal cord of humanized NSG mice (**D**). The majority of GFAP+ (red) cells were astrocytes from the mouse hosts. Data are mean ± SEM. *n* = 3. Student’s *t*-test. * *p* < 0.05; *** *p* < 0.001. Bar, 400 µm.

**Figure 9 biology-14-00791-f009:**
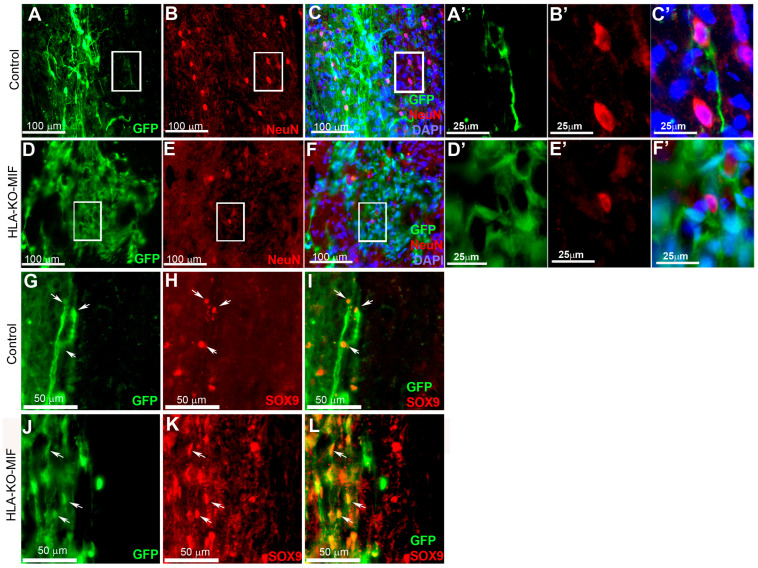
NPCs from HLA-KO-MIF iPSCs give rise to young neurons and astrocyte progenitors upon grafting to the injured spinal cord of humanized NSG mice. NPCs from NCL2-GFP control and HLA-KO-MIF iPSCs differentiated into NeuN+ young neurons upon grating to the injured spinal cord of humanized NSG mice. Only a few NeuN+ cells were detected two weeks post graft in control (**A**–**C**) and HLA-KO-MIF (**D**–**F**) groups because human NPCs usually require longer time than mouse NPCs to differentiate and mature to neurons upon grafting. (**A′**–**C′**) and (**D′**–**F′**) are higher magnification images of the boxed-in areas. (**G**–**L**) In addition, NPCs from the NCL2-GFP control and HLA-KO-MIF iPSCs (labeled by GFP) also gave rise to SOX9+ astrocyte progenitors (red) upon grafting. Arrows indicate SOX9+ astrocyte progenitors derived from hiPSCs. (**A′**–**F′**) are higher magnification images of the corresponding boxed-in areas. Bars in (**A**–**F**), 100 µm; (**A′**–**F′**), 25 µm; (**G**–**L**), 50 µm.

## Data Availability

Data are available from the authors with permission of Florida International University and University of Texas Health Science Center at Houston.

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
