# Peer review of "Macrophage Migration Inhibitory Factor Suppresses Natural Killer Cell Response and Promotes Hypoimmunogenic Stem Cell Engraftment Following Spinal Cord Injury"

_biology, 2025, doi:10.3390/biology14070791_

Round 1
Reviewer 1 Report
Comments and Suggestions for Authors
The manuscript reports on the overexpression of macrophage migration inhibitory factor (MIF) combined with knockout of B2M and CIITA to improve grafting strategies. The concept is interesting, and the results appear promising. However, several experiments lack appropriate controls, and the image quality is poor, which weakens the credibility of the manuscript. My specific comments are as follows:
- Please provide the names of the companies that supplied the chemicals used in the cell culture experiments.
- Line 133: The temperature should be written as 37°C.
- Line 139: Please describe the method used to dissociate the cells.
- In section 2.8 (Quantitative RT-PCR), please clarify which tissues were used for the qRT-PCR analysis.
- What secondary antibodies were used in the experiments? Please provide details, including the company, species, and catalog number.
- What specific proteins were each primary antibody targeting? Please specify in the methods.
- What was the purpose of IFN-γ treatment? In Line 284, the authors state: “Upon 100 ng/mL IFN-γ treatment for 48 h, expression of CIITA and HLA-II could be detected in NCL2-GFP (Figure 1D, 1E). However, the HLA-KO cells did not express CIITA and HLA-II (Figure 1D, 1E).” However, CIITA and HLA-II expression appears minimal even in NCL2-GFP. Please elaborate and clarify.
- The quality of Figure 1 needs to be improved.
- Please include a karyotype image of the parental NCL2-GFP cells in Figure 2A for comparison with the HLA-KO cells.
- Figures 2C–2O: The authors state that “Our results show that B2M/CIITA double KO does not change the properties of hiPSCs, such as karyotype, STR, or pluripotency.” However, control (non-KO) cells are not shown in the figures. Without controls, it is difficult to validate the conclusion. Please include images of control cells in all relevant experiments.
- Why do the images in Figure 2 not use the same magnification? Please standardize this for consistency.
- In Figures 2D–2O, please use arrowheads to indicate the location of specific proteins in the images.
- Why are black-and-white images used for SOX2, synaptophysin, and NF160? Please ensure consistency in image presentation.
- Please expand on the qRT-PCR results in Section 3.3, especially regarding Figure 3C.
- In Figure 4, what does the label “ns” on the bars mean? Please define this in the figure legend.
- It would be clearer to combine Figures 4A, 4B, and 4C into one composite image to better visualize the differences between conditions. Also, the Y-axis scales differ across these panels (0–50 in Fig. 4A vs. 0–25 in 4B–4D), making comparisons difficult.
- Figure 5: Please include brightfield images of the cells. It is difficult to observe cellular damage using only fluorescent images.
- Line 400: The authors mention that “These cells showed typical amoeboid morphology with an increase in cell size. In the NCL2-GFP control group (Figure 5A–5E)”. However, these features are not clearly visible in the figure, possibly due to low magnification. Please clarify.
- Please include scale bars in Figures 9A’–F’.
- Figures 9G–9N: The image quality is poor. The cells in the highlighted squares appear as mere dots. Why weren’t the cells stained with DAPI or a membrane marker? It’s unclear whether these are cells or artifacts. Brightfield images would help interpret these results.
- The manuscript contains excessive references and self-citations. Please review and reduce redundant citations.
Author Response
Point-by-point responses to the reviewers’ comments:
Reviewer 1. The manuscript reports on the overexpression of macrophage migration inhibitory factor (MIF) combined with knockout of B2M and CIITA to improve grafting strategies. The concept is interesting, and the results appear promising. However, several experiments lack appropriate controls, and the image quality is poor, which weakens the credibility of the manuscript. My specific comments are as follows:
- Please provide the names of the companies that supplied the chemicals used in the cell culture experiments.
Thank the reviewer for the suggestion. Information on vendors and suppliers of the chemicals used in cell culture experiments are provided in the Materials and Methods section. Please also note that “All reagents were from ThermoFisher unless otherwise specified” (Line 143).
2. Line 133: The temperature should be written as 37°C.
Thank the reviewer for the suggestion. This has been corrected.
3. Line 139: Please describe the method used to dissociate the cells.
Thank the reviewer for the suggestion. We have used TrypLE to dissociate the cells. This information is now provided in the Materials and Methods section, Line 141 “...by incubating in TrypLE for 5 min.”
4. In section 2.8 (Quantitative RT-PCR), please clarify which tissues were used for the qRT-PCR analysis.
Thank the reviewer for the suggestion. We have clarified the source of the materials for qRT-PCR. “Total RNAs of iPSCs or iPSC derived cells were extracted using...” (Line 223).
5. What secondary antibodies were used in the experiments? Please provide details, including the company, species, and catalog number.
Thank the reviewer for the suggestion. Information on secondary antibodies were added to the Materials and Methods section.
“Secondary antibodies used are from Jackson Immuno Research: FITC-affiniPure Don-key anti-Rabbit, Cat. No. 711-095-152; Rhd-affiniPure Donkey anti-Rabbit, Cat. No. 711-025-152; Cy5-affiniPure Donkey anti-Rabbit, Cat. No. 711-175-152; Rhd-affiniPure Donkey anti-Mouse, Cat. No. 715-025-150; Cy5-affiniPure Donkey anti-Mouse, Cat. No. 715-175-150; AMCA-affiniPure Donkey anti-Rat, Cat. No. 712-155-153; Cy5-AffiniPure Donkey Anti-Rat IgG, Cat. No. 712-175-150” (Line 250-Line 255).
6. What specific proteins were each primary antibody targeting? Please specify in the methods.
The specific proteins that each primary antibody targeting is the same as the name of the antibody, which has been listed in the Material and Methods section.
7. What was the purpose of IFN-γ treatment? In Line 284, the authors state: “Upon 100 ng/mL IFN-γ treatment for 48 h, expression of CIITA and HLA-II could be detected in NCL2-GFP (Figure 1D, 1E). However, the HLA-KO cells did not express CIITA and HLA-II (Figure 1D, 1E).” However, CIITA and HLA-II expression appears minimal even in NCL2-GFP. Please elaborate and clarify.
Thank the reviewer for the question. CIITA and HLA-II are not expressed under normal conditions; however, their expression is upregulated following IFN-γ treatment. Therefore, we used IFN-γ to induce their expression. In the HLA-KO cells, CIITA and HLA-II expression remained absent even after IFN-γ treatment, confirming the complete loss of HLA-II expression.
8. The quality of Figure 1 needs to be improved.
Thank the reviewer for the suggestion. Figure 1A presents the chromatogram from sequencing results. Figures 1B–E were captured using a Zeiss Axiovision microscope with the Z-stack function. To visualize gene expression across a broader field and include as many cells as possible, we selected images taken with a 20x objective lens.
9. Please include a karyotype image of the parental NCL2-GFP cells in Figure 2A for comparison with the HLA-KO cells.
Thank the reviewer for the suggestion. The parental NCL2-GFP cells were previously described in Reference 23. To avoid duplication, we cited the original publication in our manuscript but did not include data already presented in that study.
Reference No. 23: Rao, M.S.; Pei, Y.; Garcia, T.Y.; Chew, S.; Kasai, T.; Hisai, T.; Taniguchi, H.; Takebe, T.; Lamba, D.A.; Zeng, X. Illustrating the potency of current Good Manufacturing Practice-compliant induced pluripotent stem cell lines as a source of multiple cell lineages using standardized protocols. Cytotherapy 2018, 20, 861-872, doi:10.1016/j.jcyt.2018.03.037.
10. Figures 2C–2O: The authors state that “Our results show that B2M/CIITA double KO does not change the properties of hiPSCs, such as karyotype, STR, or pluripotency.” However, control (non-KO) cells are not shown in the figures. Without controls, it is difficult to validate the conclusion. Please include images of control cells in all relevant experiments.
Thank the reviewer for the suggestion. As shown in our manuscript, the control (non-KO) cells are NCL2-GFP. This cell line was previously described in Ref#23. To avoid duplication, we cited the original publication in our manuscript but did not include data already presented in that study.
11. Why do the images in Figure 2 not use the same magnification? Please standardize this for consistency.
Thank the reviewer for the suggestion. Figures 2C–2F display pluripotency markers, while Figures 2G–2O show differentiation markers. Each panel is intended to clearly demonstrate the expression of a specific marker. As such, magnifications may vary between panels; however, each image includes its own scale bar to indicate the magnification used. This variation does not compromise the clarity or interpretation of the data.
12. In Figures 2D–2O, please use arrowheads to indicate the location of specific proteins in the images.
Thank the reviewer for the suggestion. Adding arrowheads would obscure portions of the immunostaining signals. As the staining is clearly labeled, readers should be able to readily identify the positively stained cells and the corresponding marker expression without additional annotations.
13. Why are black-and-white images used for SOX2, synaptophysin, and NF160? Please ensure consistency in image presentation.
The white signal on a black background represents the pseudocolor scheme applied during image acquisition.
14. Please expand on the qRT-PCR results in Section 3.3, especially regarding Figure 3C.
Thank the reviewer for the suggestion. We have provided additional information in Figure legend for Figure 3C
“Data are mean ± SEM. n = 3. Student's t-test. * P < 0.05; **P < 0.01; *** P < 0.001. ns, not significant.” (Line 375 to Line 376)
15. In Figure 4, what does the label “ns” on the bars mean? Please define this in the figure legend.
Thank the reviewer for the suggestion. We have provided additional information in Figure legend for Figure 4:“ns, not significant.” (Line 389 to Line 390)
16. It would be clearer to combine Figures 4A, 4B, and 4C into one composite image to better visualize the differences between conditions. Also, the Y-axis scales differ across these panels (0–50 in Fig. 4A vs. 0–25 in 4B–4D), making comparisons difficult.
Thank the reviewer for the suggestion. The purpose of the graphs in Figures 4A–4C is to compare NK cell cytotoxicity among the three groups at each E:T ratio, rather than to compare across different E:T ratios. Figure 4D presents data from a separate assay measuring the percentage of NK cell-mediated target cell degradation.
17. Figure 5: Please include brightfield images of the cells. It is difficult to observe cellular damage using only fluorescent images.
The purpose of Figure 5 is not to show cellular damage, but to show the inflammatory cell response at the injury site.
18. Line 400: The authors mention that “These cells showed typical amoeboid morphology with an increase in cell size. In the NCL2-GFP control group (Figure 5A–5E)”. However, these features are not clearly visible in the figure, possibly due to low magnification. Please clarify.
Thank the reviewer for the suggestion. Line 400 (now Line 409-410) has been changed to “These cells showed the typical amoeboid morphology”.
19. Please include scale bars in Figures 9A’–F’.
Thank the reviewer for the suggestion. We have added scale bars for Figure 9A′–9F′, and added information in Figure 9 legend (Line 491).
20. Figures 9G–9N: The image quality is poor. The cells in the highlighted squares appear as mere dots. Why weren’t the cells stained with DAPI or a membrane marker? It’s unclear whether these are cells or artifacts. Brightfield images would help interpret these results.
The highlighted squares indicate examples of positively stained cells. All available fluorescence channels were utilized in these images: GFP (green), human nuclei (hN, red), SOX9 (white, using the far-red channel), and GFAP (blue). As a result, DAPI staining was not performed.
21. The manuscript contains excessive references and self-citations. Please review and reduce redundant citations.
The citations and references, including those to our own prior work, form the foundation of this manuscript. We have carefully reviewed and confirmed that all referenced works are essential and relevant to the content presented.
Reviewer 2 Report
Comments and Suggestions for Authors
The authors have presented a very serious, interesting and well-illustrated study.
1) Have cell cultures been tested for mycoplasma contamination? How?
2) Figure 3 is mentioned before all other figures. Inappropriate. The authors should place the construct diagram as Figure 1.
3) The figures should follow in the text immediately after the first mention.
4) As a result of various injuries, astrocyte reactivation can occur, which can be of type A1 and A2. Is reactivation of astrocytes possible in the experiments presented in the article?
5) The authors should structure the discussion
Author Response
Reviewer 2
Comments and Suggestions for Authors
The authors have presented a very serious, interesting and well-illustrated study.
1) Have cell cultures been tested for mycoplasma contamination? How?
Thank the reviewer for the suggestion. We have added information regarding mycoplasma contamination in the Materials and Methods section.
“All cell cultures were routinely tested for mycoplasma contamination on a monthly basis using a Mycoplasma PCR detection kit (ABM, Cat. No. G238).” (Line 105-107)
2) Figure 3 is mentioned before all other figures. Inappropriate. The authors should place the construct diagram as Figure 1.
Thank the reviewer for the suggestion. We have revised the manuscript to ensure that all figures appear in the correct order within the text.
3) The figures should follow in the text immediately after the first mention.
Thank the reviewer for the suggestion. Due to layout and typographic design, some figures do not appear immediately after their corresponding text. We will work with the editor to ensure proper placement in the final version.
4) As a result of various injuries, astrocyte reactivation can occur, which can be of type A1 and A2. Is reactivation of astrocytes possible in the experiments presented in the article?
Thank the reviewer for the suggestion. Astrocyte reactivation does occur following spinal cord injury. However, as the primary focus of this study is on hypoimmunogenic cells, this aspect was not discussed in detail.
5) The authors should structure the discussion
Thank the reviewer for the suggestion. In the Discussion section, we summarized our findings and described our strategy for generating hypoimmunogenic iPSCs, with a focus on using MIF to prevent NK cell–mediated attack. We also acknowledged the limitations of using a humanized model to evaluate this approach in the context of spinal cord injury.
Round 2
Reviewer 1 Report
Comments and Suggestions for Authors
The authors have responded to the comments, and some of them have been addressed through revisions. In certain cases, the authors explained their reasons for not making changes. However, many issues remain unclear.
For example, the authors declined to add a control to Figure 2, stating that the control cell line (NCL2-GFP) was previously published. This is not a sufficient justification. Did you use NCL2-GFP in this experiment? If so, an image of the control from this experiment should be included. Using it as a control in this context would not constitute duplication. However, if no control image is provided, the results are not strongly supported.
My comments aim to help improve the manuscript so that readers can better understand the findings. While the results are promising, the manuscript remains difficult to follow.
Regarding Figures 9G–N, the cells or tissues in the control group appear different from those in the HLA-KO-MF group. Specifically, the cells in the HLA-KO-MF group (Figure 9K) appear larger than those in the control group (Figure 9G). I requested brightfield images to provide a clearer comparison, but the authors declined.
In my opinion, readers may experience the same confusion. I strongly believe the manuscript would benefit from revisions to enhance clarity and readability.
Author Response
Comments: The authors have responded to the comments, and some of them have been addressed through revisions. In certain cases, the authors explained their reasons for not making changes. However, many issues remain unclear.
For example, the authors declined to add a control to Figure 2, stating that the control cell line (NCL2-GFP) was previously published. This is not a sufficient justification. Did you use NCL2-GFP in this experiment? If so, an image of the control from this experiment should be included. Using it as a control in this context would not constitute duplication. However, if no control image is provided, the results are not strongly supported.
My comments aim to help improve the manuscript so that readers can better understand the findings. While the results are promising, the manuscript remains difficult to follow.
Response: Thank the reviewer for the suggestion. We have included a karyotype image of our control line NCL2-GFP in Supplementary Figure 1.
Comments: Regarding Figures 9G–N, the cells or tissues in the control group appear different from those in the HLA-KO-MF group. Specifically, the cells in the HLA-KO-MF group (Figure 9K) appear larger than those in the control group (Figure 9G). I requested brightfield images to provide a clearer comparison, but the authors declined.
In my opinion, readers may experience the same confusion. I strongly believe the manuscript would benefit from revisions to enhance clarity and readability.
Response: We thank the reviewer for the suggestion. We have reassembled Figure 9G-N (now revised as Figure 9G-L) to include higher magnification images, allowing for a clearer comparison of astrocyte progenitor (SOX9+) cell differentiation between control and HLA-KO-MIF lines.
Reviewer 2 Report
Comments and Suggestions for Authors
The authors took into account all my comments. I recommend the article for publication in its current form.
Author Response
Comments: The authors took into account all my comments. I recommend the article for publication in its current form.
Response: Thank the reviewer for their time and positive review.